# Degranulation of Mast Cells as a Target for Drug Development

**DOI:** 10.3390/cells12111506

**Published:** 2023-05-29

**Authors:** Bo-Gie Yang, A-Ram Kim, Dajeong Lee, Seong Beom An, Yaein Amy Shim, Myoung Ho Jang

**Affiliations:** 1Research Institute, GI Biome Inc., Seongnam 13201, Republic of Korea; arkim@gi-biome.com (A.-R.K.); yodj.lee@gi-biome.com (D.L.); sban@gi-biome.com (S.B.A.); 2Research Institute, GI Innovation Inc., Songpa-gu, Seoul 05855, Republic of Korea; yaein.shim@gi-innovation.com

**Keywords:** mast cells, degranulation, IgE-targeted drugs

## Abstract

Mast cells act as key effector cells of inflammatory responses through degranulation. Mast cell degranulation is induced by the activation of cell surface receptors, such as FcεRI, MRGPRX2/B2, and P2RX7. Each receptor, except FcεRI, varies in its expression pattern depending on the tissue, which contributes to their differing involvement in inflammatory responses depending on the site of occurrence. Focusing on the mechanism of allergic inflammatory responses by mast cells, this review will describe newly identified mast cell receptors in terms of their involvement in degranulation induction and patterns of tissue-specific expression. In addition, new drugs targeting mast cell degranulation for the treatment of allergy-related diseases will be introduced.

## 1. Introduction

Although mast cells are well known for their role as essential effectors involved in allergic reactions, they are difficult to find in the blood because mast cell progenitors (MCPs) migrate to tissues before maturing into mast cells [1,2]. In addition, MCPs differentiate and mature into mast cells with heterogeneous phenotypes based on the tissue environment. The traditional way of dividing mast cells into subtypes differs between mice and humans. Murine mast cells can be divided into mucosal mast cells (MMCs) or connective tissue mast cells (CTMCs) [2,3,4]. MMCs, present in the mucosal epithelium of the lung and gastrointestinal tract, express mouse mast cell protease (mMCPT)-1 and -2. Upon activation, MMCs release mostly leukotrienes with a small amount of histamine [4]. In contrast, CTMCs, which are present in the intestinal submucosa, peritoneum, and skin, express mMCPT-4, -5, -6, and carboxypeptidase A (CPA) and release high levels of histamine and prostaglandin D2 [4]. Human mast cells are classified based on the pattern of protease expression. MC_T_ predominantly expresses tryptase and shows similar characteristics to murine MMC. MC_TC_ expresses tryptase and chymase and is more similar to murine CTMC [3,4]. Under homeostatic conditions, the number of mast cells in vivo is meager. Upon encountering stimuli, mast cell activation occurs leading to proliferation and degranulation, inducing the onset of allergy. Mast cell degranulation is generally known to occur when IgE binds to the high-affinity IgE receptor FcεRI, but recently, other receptors involved in IgE-independent degranulation of mast cells have been described. Unlike FcεRI, the receptors involved in IgE-independent degranulation are not expressed in all mast cells. Their expression patterns vary depending on the tissue [5], allowing for the subtyping of tissue-specific mast cells based on the expression pattern of the receptors. This review focuses on mast cell receptors which activate degranulation and inflammation, as well as mechanisms for the activation and proliferation of mast cells. It also introduces ongoing efforts to translate insights from mast cell degranulation into the development of biological drugs for the treatment of allergic diseases.

## 2. Factors Involved in the Differentiation, Proliferation, and Activation of Mast Cells

### 2.1. SCF (Stem Cell Factor) and IL-3

Mast cells are commonly characterized by the expression of SCF receptor KIT and a high-affinity IgE receptor FcεRI [2]. Most MCPs also express these two receptors, but unlike mature mast cells, they do not have granules in the cytoplasm [4]. When cultured in the presence of SCF and IL-3, mouse MCPs mature into mast cells with increased granule formation and FcεRI expression [3,4]. SCF plays a critical role in mast cell differentiation, proliferation, and survival, as shown by the mast cell deficiency in mice with a loss-of-function mutation in the genes of SCF or KIT [3]. In addition to mast cells, SCF participates in the development and function of multiple distinct cell lineages including hematopoietic progenitors, melanocytes, and germ cells [6]. SCF exists in both membrane-bound and soluble forms, and the SCF receptor KIT is activated only when SCF is bound as a homodimer rather than a monomer [7]. Membrane and soluble SCF each seem to have different biological functions in hematopoiesis, and soluble SCF is more important in mast cell development and survival [8,9]. In support of this finding, membrane SCF embedded in proteoliposomes or lipid nanodiscs improved revascularization in ischemic hind limbs without activating mast cells [10]. However, soluble SCF variants engineered to impair KIT dimerization, thereby reducing the downstream signaling amplitude, induced biased activation only in hematopoietic progenitors, not mast cells in vitro and in vivo [11]. In addition, mast cell degranulation caused by antigen-induced aggregation of the IgE–FcεRI complex is further increased by the binding of SCF to the receptor KIT [12,13]. This synergistic effect occurs as both SCF- and antigen-induced mast cell activation each have their own signaling pathway, and convergent signaling is achieved by non-T cell activation linker (NTAL) phosphorylation [13]. The microbiota signal keratinocytes to produce SCF, which induces the recruitment and maturation of skin mast cells [14]. For example, germ-free (GF) mice have a very a low SCF level and have mostly undifferentiated mast cells in the skin and intestine [14,15,16]. Accordingly, GF mice were less prone to develop a food allergy than specific-pathogen-free (SPF) mice [16]. However, GF mice were previously reported to have a higher level of IgE than SPF mice, with greater susceptibility to a food allergy model [17,18]. The discrepancy between these experimental results was speculated to be from the difference in the food allergy protocols, including the sterilization methods used to prepare chow [16].

Similar to SCF, adding IL-3 to the culture can also enhance mast cell survival, development, and maturation [4,19]. However, the characterization of IL-3-deficient mice affirmed IL-3 is not essential for the generation of mast cells under physiological conditions but contributes to the increase of mast cells in the presence of parasitic infection [20]. The role of IL-3 in the development and maintenance of mast cells in humans has conflicting reports and remains to be elucidated [4,19].

### 2.2. IL-4 and IL-33

On top of IgE production and Th2 cell differentiation, IL-4 also plays a vital role in inducing the proliferation of mast cells [21]. For example, intestinal mast cells were increased when mice were administered with IL-4 [22]. In addition, mice expressing *Il-4rαF709* in all cells, a variant of the IL-4 receptor α chain (IL-4Rα) with enhanced IL-4R signaling, have an increased number of intestinal mast cells and show more severe anaphylactic reactions to the food allergy model compared to wild-type mice [22,23]. Furthermore, it was confirmed that IL-4 is directly involved in the expansion of mast cells in vivo [22]. The study used a mouse model whose endogenous MCPs had been removed by sublethal irradiation and reconstituted with a 50:50 mixture of bone marrow from CD45.1^+^ wild-type and CD45.2^+^ *IL-4Rα^−/−^* mice. The mesenteric lymph node (MLN) cells from ovalbumin (OVA)-sensitized *Il-4rαF709* mice were adoptively transferred and then fed with OVA on days 1, 4, and 7 from the reconstitution. On days 10–11, the number of CD45.2^+^ *IL-4Rα^−/−^* mast cells was significantly less than the number of CD45.1^+^ wild-type mast cells, indicating that IL-4 signaling is important for mast cell expansion. In addition, IL-4 enhances IgE-mediated mast cell responses such as FcεRI expression, cytokine production, and degranulation [24]. However, other studies report that IL-4-induced mast cell apoptosis reduced the surface expression of FcεRI and KIT [24]. Such discrepancies may have come from differences in the degree of the maturation of the mast cells, and the IL-4 may play a different role in mature versus immature mast cells. Despite the controversies in reports on the role of IL-4 in mast cells, the general understanding of IL-4 is that it is an essential cytokine that can directly act on mast cells, inducing proliferation and enhancing IgE-mediated responses.

MCPs and mast cells constitutively express suppression of tumorigenicity 2 (ST2; also known as IL-1RL1, T1, and IL-33R) [25,26]. ST2 forms a heterodimer with the IL-1 receptor accessory protein (IL-1RAcP; also known as IL1RAP) and exerts biological effects by binding to IL-33. IL-33, similar to IL-25 and thymic stromal lymphopoietin (TSLP), is an alarmin secreted from epithelial barrier cells when damaged by external stimuli from exposure to trauma, infection, or allergen [27]. The binding of IL-33 to ST2 on human mast cells promotes the secretion of cytokines, such as IL-5 and IL-13, and chemokines, such as CXCL8 and CCL1, and increases mast cell survival by suppressing apoptosis [28,29]. IL-33 is a particularly potent activator of type 2 innate lymphoid cells (ILC2s), which produce Th2 cytokines such as IL-4, IL-5, and IL-13 [27,30]. Thus, when administered to naive mice, IL-33 increases the expression of Th2 cytokines (e.g., IL-4, IL-5, and IL-13) in tissues and IgE levels in blood and causes eosinophilia [31]. The increase in serum IgE levels and eosinophil number is an indirect effect of IL-4 and IL-5, respectively, rather than a direct effect of IL-33 [32,33]. Moreover, IL-33 was shown to trigger anaphylaxis in an IL-4-dependent manner in naive mice [33]. IL-33 contributes to type 2 inflammation by acting on cells other than mast cells. Namely, Th2 cells and group 2 innate lymphoid cells strongly respond to IL-33 through ST2 [26]. In one study, the contribution of TSLP, IL-25, and IL-33 in the development of allergic diseases was compared in various sensitization models using neutralizing antibodies and knockout mouse models. Interestingly, only IL-33 signaling was shown to be necessary for mite and peanut allergic sensitization [34]. TSLP alone has little effect on mast cells but enhances IL-33-induced cytokine secretion by mast cells and MCPs [29]. Additionally, allergen sensitization, either through the skin or intestine, induces mast cell activation in an IgE-dependent manner, causing oral allergen-induced anaphylaxis [35,36,37]. Particularly, food allergen sensitization through the skin increases local and systemic levels of IL-33, ultimately inducing the expansion of mast cells in the intestine [38]. However, the expansion of mast cells by IL-33 is achieved indirectly by inducing IL-4 secretion of ILC2 rather than by a direct action of IL-33 [30]. Intestinal ILC2 activated by IL-33 produces not only IL-4 but also IL-13. In addition, IL-33, along with IL-25 from the tuft cells stimulated with ILC2-derived IL-13, induces the activation and expansion of ILC2, and enhances IL-4 production [30]. Mast cells can also produce IL-33 upon IgE stimulation, as well as P2RX7 activation by helminth infection [39,40]. The blocking of IgE in a food allergy mouse model reduces *Il-33* expression and the number of intestinal mast cells [37].

### 2.3. IL-9

IL-9-deficient and IL-9-overexpressing mice helped discover the central role of IL-9 in mast cell proliferation and goblet cell hyperplasia [41,42]. IL-9 acts directly on mast cells to promote their proliferation in vitro [43]. IL-9 is produced not only in ILC2, Th2, and IL-9-producing CD4^+^ T (Th9) cells but also in mast cells and is involved in both the activation and proliferation of mast cells [44]. While IL-9-deficient mice fail to develop oral antigen-induced anaphylaxis, the overexpression of intestinal IL-9 induces an anaphylactic phenotype [45]. In the same study, IL-9 was described to promote oral antigen sensitization by increasing mast cells in the small intestine and intestinal permeability. However, IL-9 signaling plays an important role only in oral antigen-induced anaphylaxis, not in parenteral antigen-induced systemic anaphylaxis [46]. IL-9-producing mucosal mast cells (MMC9s) characterized by Lin^-^IL-17RB^-^ KIT^+^ST2^+^β7integrin^lo^, as well as MCPs by Lin^-^IL-17RB^-^ KIT^+^ST2^+^β7integrin^hi^ phenotype, can be identified in the mouse intestine [47]. MMC9 induction occurs upon a repeated oral antigen challenge, and this process requires IL-4 secretion from Th2 cells, while IL-9 mediates the proliferation of MMC9s in an autocrine manner [45]. The blocking of α4β7 integrin, which drives the intestinal migration of bone marrow MCPs, significantly reduced the number of MCPs and MMC9s in the intestine, suggesting that bone marrow MCPs give rise to MMC9s in the intestine [47]. When these two populations are cultured in the presence of SCF and IL-3, MMP9s seem to become less mature than MCPs [48]. When stimulated with IL-33, MMC9s increase the production of IL-9 and IL-13 and play a critical role in the pathogenesis of the IgE-mediated food allergy [47].

## 3. Receptors Involved in Mast Cell Activation and Degranulation

### 3.1. High-Affinity IgE Receptor (FcεRI)

IgE plays an important role in allergic reactions and is involved in the pathogenesis of allergic diseases. Mast cells, which express the high-affinity IgE receptor FcεRI, are major effector cells for allergy responses and are degranulated by IgE. FcεRI exists as heterotetramer αβγ2 and heterotrimer αγ2 in humans [49]. In particular, the α chain with the extracellular domain is involved in IgE binding, while the β and γ chains are involved in signal transduction. FcεRI binding of the IgE–FcεRI complex cross-linked by an allergen induces degranulation signals to the mast cells [49]. In the absence of an allergen, FcεRI binds monomeric IgE, which provides anti-apoptosis signals to the mast cells, increasing mast cell survival [50,51,52]. For example, mice transplanted with IgE-producing hybridomas have increased gastric mast cells and blood IgE levels [50]. IgE has also been shown to increase FcεRI expression in mast cells both in vitro and in vivo [53]. Peritoneal mast cells of IgE-deficient mice show lower FcεRI expression than wild-type mice under a homeostatic condition [53]. In the food allergy model, FcεRI signaling in mast cells is essential for food allergen sensitization [36]. The Treg cells from a food allergy mouse model can be reprogramed with a Th2-cell-like phenotype and are less efficient at suppressing mast cell activity and proliferation [54]. However, inhibiting IgE signals during allergen ingestion was shown to reverse the established food allergy and restore Treg cell induction, suggesting that IgE plays a crucial role in food allergies [36].

Currently, the most promising method for food allergy treatment is oral immunotherapy (OIT), which aims to induce tolerance to food allergens by gradually raising food allergen intake starting from a very small amount [55]. In 2020, peanut allergen powder was approved by the U.S. Food and Drug Administration (FDA) as a first-in-class drug for peanut allergy [56]. However, the potential benefit of OIT is accompanied by a severe and life-threatening risk of anaphylaxis [55]. To mitigate the risk of OIT, a combination therapy with an anti-IgE antibody (Ab) has been explored. A pilot study conducted in children with peanut allergy showed that adding anti-IgE Ab to OIT significantly reduced the frequency and severity of the unwanted allergic reaction to the OIT, allowing for higher dosing of the allergen [57]. Closer examination showed that the anti-IgE Ab reversed the Th2-cell-like phenotype of Treg cells and restored normal function [58].

### 3.2. Low-Affinity IgG Receptors (FcγRs)

Although anaphylaxis is mostly associated with the cross-linking of FcεRI-bound IgE by antigens, the cross-linking of low-affinity FcγRs by antigen-bound IgG can also lead to anaphylaxis [59]. IgG-mediated systemic anaphylaxis in mice can be mediated largely through IgG1 and FcγRIII [60]. As FcγRIII is an activating receptor whereas FcγRIIB is an inhibitory receptor, IgG-mediated passive systemic anaphylaxis (PSA) is reduced in *FcRIII^−^^/−^* mice and is enhanced in *FcRIIB^−^^/−^* mice [61]. Indeed, mouse mast cells express FcγRIII and FcγRIIB, as well as FcεRI [62], and the absence of FcεRIα enhances FcγRIII-dependent mast cell degranulation and anaphylaxis [63]. Human mast cells express FcεRI and FcγRIIA, but unlike human basophils, human mast cells do not express FcγRIIB [62]. Humanized mice expressing human low-affinity IgG receptors comprising both activating (FcγRIIA, FcγRIIIA, and FcγRIIIB) and inhibitory (FcγRIIB) FcγRs are susceptible to PSA [64]. In these mice, FcγRIIA has been shown to play a predominant role in inducing anaphylaxis.

Platelet-activating factor (PAF) secreted during degranulation plays an important role in anaphylactic reactions [59]. In humans, PAF levels in the plasma increase significantly after an anaphylactic reaction and correlate strongly with the severity of the response [65]. Mast cells, neutrophils, and macrophages are some of the cell populations that produce PAF, and interestingly, they can also respond to PAF [59]. Human lung mast cells and peripheral blood (PB)-derived mast cells express the PAF receptor, but human skin mast cells do not [66]. Notably, PAF induces histamine release from human lung mast cells and PB-derived mast cells but not from skin mast cells. These findings suggest that PAF enhances mast cell activation and degranulation during anaphylaxis, leading to the aggravation of symptoms.

### 3.3. MRGPRX2/B2

Mast cell degranulation in an IgE-independent hypersensitivity reaction is activated through Mas-related G-protein-coupled receptor X2 (MRGPRX2), not FcεRI [67,68]. While FcεRI is expressed in all mast cells, MRGPRX2 is preferentially expressed in MC_TC_ in skin tissue [5,67,68]. In mice, MRGPRB2, the mouse ortholog of MRGPRX2, is predominantly expressed in CTMCs [69], which are the mouse equivalent of MC_TC_. MRGPRX2/B2 binds to various agonists, including insect venom (e.g., Mastoparan), chemical components (e.g., Compound 48/80, opioids, etc.), antimicrobial peptides (e.g., LL-37, β-defensins, etc.), neuropeptides (e.g., Cortistatin-14 (CST-14), substance P (SP), etc.), and FDA-approved drugs (e.g., Icatibant, Cetrorelix, Leuprolide, Octreotide, Sermorelin, Atracurium, Tubocurarine, Rocuronium, Ciprofloxacin, etc.) [67,69]. The homology in the amino acid sequence between MRGPRX2 and MRGPRB2 is only about 53%, and the two receptors differ in the concentration of agonists required for activation, as well as in their ligand selectivity for activation and inhibition [70]. MRGPRX2/B2 is involved in mast-cell-mediated host defense against bacterial infection. Mast cells sense bacterial infection when MRGPRX2/B2 binds quorum-sensing molecules (QSMs) secreted from bacteria or antimicrobial host defense peptides (HDPs) generated from infection-activated keratinocytes [68,71]. In addition, it has been reported that mast cells contribute to atopic dermatitis, allergic contact dermatitis, nonhistaminergic itch, neurogenic inflammation and pain, and pseudo-allergic responses through MRGPRB2 in mouse models [68]. In an atopic dermatitis mouse model, the house dust mite (HDM) *Dermatophagoides farinae* stimulates transient receptor potential vanilloid type 1 (TRPV1^+^) sensory neurons in the skin to secrete SP, which triggers the activation and degranulation of the nearby mast cells by binding to MRGPRB2, thereby inducing atopic dermatitis (Figure 1) [72]. Thus, mice deficient in the SP gene *Tac1* and those treated with resiniferatoxin for selective removal of the TRPV1^+^ nociceptor show alleviated symptoms of atopic dermatitis. Similarly, mast-cell-deficient and *Mrgprb2^Mut^* mice show the same phenomenon [72]. In the skin of atopic dermatitis patients, an increase in the number of mast cells and signs of mast cell degranulation are observed near SP-expressing nerve bundles [73,74]. Moreover, the number of mast cells and expression of MRGPRX2 ligand PAMP1-20 were increased in the damaged skin of patients with allergic contact dermatitis, a pruritic skin disease caused by repetitive hapten exposure [75]. The activation of MRGPRB2 in mouse mast cells induces pruritus in an IgE-independent manner. Unlike IgE-induced pruritus, which is caused by the activation of itch sensory neurons by histamine and serotonin released from degranulation, MRGPRB2-induced pruritus is caused by the activation of itch sensory neurons by tryptase (Figure 1) [75]. Therefore, anti-histamine treatment was ineffective against pruritus mediated by MRGPRB2. In the postoperative model of inflammatory pain, the activation of mast cell MRGPRB2 induced the secretion of cytokines and chemokines, which leads to the recruitment of immune cells, causing neurogenic inflammation and pain [76]. Furthermore, most FDA-approved drugs associated with allergic-type injection-site reactions contribute to the induction of systemic pseudo-allergic and anaphylactoid reactions through the activation of MRGPRB2 [69]. In summary, mast cells and MRGPRX2/B2 are closely involved in IgE-independent hypersensitivity reactions.

### 3.4. P2X Purinoceptor 7 (P2RX7)

Unlike skin mast cells, intestinal mast cells preferentially express the extracellular ATP receptor P2RX7 and participate in inflammatory responses [77,78]. ATPs are released into the extracellular space when cells are damaged and act as a danger signal to mast cells [79]. P2RX7 expression in mast cells is regulated by retinoic acid, and retinoic acid level is regulated by a degrading enzyme, CYP26B1, whose expression is high in skin fibroblasts (Figure 1) [77]. Thus, the mast cell expression of P2RX7 is suppressed in the skin but is strong in the intestine under homeostatic conditions. However, the inhibition of CYP26B1 receptor activity increases P2RX7 expression in skin mast cells [77]. Likewise, high concentrations of retinoic acid also increase P2RX7 expression in skin mast cells and cause retinoid dermatitis [77]. Intestinal mast cells produce a large amount of IL-33 in response to extracellular ATPs, which induces the activation and proliferation of ILC2 [40]. IL-13 secreted by ILC2 induces goblet cell hyperplasia for parasite expulsion during a parasitic infection, such as *Heligmosomoides polygyrus* [40]. In addition, intestinal mast cells activated by P2RX7 secrete cytokines, chemokines, and leukotrienes, marking the initiation of intestinal inflammation (Figure 1) [78]. Also expressed by macrophages, dendritic cells, and enteric neurons, P2RX7 is involved in the inflammatory responses of a variety of cells [79].

### 3.5. Adhesion G-Protein-Coupled Receptor E2 (ADGRE2)

ADGRE2, also known as EGF-like module-containing mucin-like hormone receptor-like 2 (EMR2) or CD132, belongs to an adhesion G-protein-coupled receptor family [80]. It consists of a large extracellular domain (α subunit) and a seven-transmembrane domain (β subunit). Although the α and β subunits of ADGRE2 are initially translated into a single polypeptide precursor, they are non-covalently linked by autocatalytic cleavage [80]. ADGRE2 is primarily expressed in myeloid leukocytes [81] but also on the surface of lung mast cells and the HMC1 human mast cell line [82].

The endogenous ligand of ADGRE2 is dermatan sulfate, which is the predominant glycosaminoglycan in the skin [80]. Despite the binding of ADGRE2 to dermatan sulfate itself not inducing a detectable mast cell activation response, mechanical force added to dermatan sulfate binding induces mast cell degranulation [80]. When mast cells attached to dermatan sulfate are stimulated by mechanical vibration, the α subunit of ADGRE2 dissociates from the β subunit, leading to mast cell activation [83]. Thus, mechanical forces seem to activate ADGRE2 by separating the α and β subunits. The p.C492Y mutation in ADGRE2, which replaces cysteine with tyrosine at amino acid 492, is found in patients with vibratory urticaria. This mutation destabilizes the inhibitory interaction between the α and β subunits, increasing the susceptibility of the mast cells to vibration-induced degranulation [83]. In addition, α/β-tryptase heterotetramers make mast cells susceptible to vibration-induced degranulation by cleaving the α subunit of ADGRE2 [84]. Therefore, targeted inhibitors of α/β-tryptase may have clinical utility.

## 4. Development of Drugs Targeting the Inhibition of Mast Cell Degranulation

### 4.1. Inhibition of IgE-Dependent Degranulation

Most early biological drugs developed for allergy treatment were anti-IgE Abs that inhibit the binding of IgE to FcεRI on the effector cells. Efforts have been made to strategically block the binding of IgE to FcεRI without affecting the binding of IgE to the low-affinity IgE receptor CD23 to increase the treatment’s potency. CD23 is expressed in a variety of cells, including T cells, B cells, polymorphonuclear leucocytes, monocytes, follicular dendritic cells, intestinal epithelial cells, and bone marrow stromal cells [85]. In B cells, CD23 is involved in the suppression of IgE production [85]. Presently, a fusion protein that links the FcεRIα domain to the Fc domain of Abs is being developed as an IgE blocking agent.

Various types of biologic drugs have been developed, but many have been discontinued for various reasons. For example, quilizumab is a humanized monoclonal Ab (mAb) that recognizes only membrane IgE (mIgE), not serum IgE. It has an afucosylated Fc domain to enhance its binding to FcγRIIIA and functions by eliminating mIgE-positive B cells via NK-cell-mediated antibody-dependent cellular cytotoxicity (ADCC) [86,87]. However, in phase 2 clinical trials with patients with allergic asthma or chronic spontaneous urticaria (CSU), quilizumab only partially reduced serum IgE levels and did not significantly improve symptoms. Compared to omalizumab, XmAb7195 is an anti-IgE mAb with 5.3-fold higher IgE binding affinity and 400-fold higher binding affinity to the inhibitory IgG receptor FcγRIIB due to two point mutations in the IgG1 Fc domain [86,87]. It functions by blocking free IgE and inhibiting IgE production through the aggregation of mIgE and FcγRIIB in B cells. In a phase 1a clinical trial with healthy adult volunteers with elevated IgE levels, XmAb7195 decreased IgE levels. However, the result of its phase 2b clinical trial has not been published since its completion in 2017. MEDI4121 is a human IgG1 anti-IgE antibody with an IgE binding affinity 100 times higher than that of omalizumab and an increased binding ability to FcγRIIIA due to an insert mutation of the Fc domain [86,87]. Thus, MEDI4121 not only blocks IgE but also kills mIgE-positive B cells by NK-cell-mediated ADCC. However, in a phase 1 clinical trial with atopic subjects, the recovery of free IgE to the baseline was much faster in subjects treated with MEDI4121 than with omalizumab. In addition, efforts have been made to block IgE using designed ankyrin repeat proteins (DARPins), which are engineered small proteins that can recognize targets with high specificity. DARPin E2_79 can significantly inhibit IgE binding to FcεRI and actively removes IgE already bound to FcεRI [86,87]. Combining DARPin E2_79 with non-disruptive DARPin E3_53, which recognized FcεRI-bound IgE, resulted in DARPin bi53_79, which had enhanced disruptive efficacy [86,87]. However, a clinical trial has not yet been conducted. GE2 is a Fcγ–Fcε functional protein in which the Fc domain of human IgG1 (Cγ2-3) and the human Fc domain of IgE (Cε2-4) are linked [86,87]. It inhibits the degranulation of effector cells such as mast cells by aggregating FcεRI and FcγRIIB and suppresses IgE production in B cells by the aggregation of CD23 and FcγRIIB. However, a clinical trial for GE2 has not been conducted due to the production of anti-drug Abs resulting in acute anaphylaxis-like reactions in monkeys after receiving repeated injections. Single domain antibodies (sdabs), also known as nanobodies, are antibody fragments composed of a single monomeric variable Ab domain. The 026 sdab binds to the Cε3 and Cε4 domains of human IgE and inhibits IgE–CD23 binding [86,87]. Since 026 sdab causes a closed conformation change in IgE upon binding, it prevents the interaction of IgE with FcεRI and CD23 due to steric hindrance. However, a clinical trial has not yet been conducted, possibly due to the more recent development strategy preferring the conservation of IgE interaction with CD23.

#### 4.1.1. Omalizumab

Omalizumab (Xolair) is the only anti-IgE agent approved by the FDA to date. It is a humanized IgG1 mAb that blocks IgE binding to FcεRI by recognizing the Cε3 domains of IgE [86]. Omalizumab improves allergic symptoms by reducing the free IgE levels in the blood and FcεRI expression on effector cells. However, injection site reactions and systemic anaphylaxis have been reported from the clinical trials of omalizumab [88,89,90]. More recently, a study on omalizumab reported that these adverse events occur through the interaction between the IgG1 Fc domain and IgG receptor FcγR using an FcγR-humanized mouse model expressing hFcγRI, hFcγRIIa_H131_, hFcγRIIb_I232_, hFcγRIIc_stop13_, hFcγRIIIa_v158_, and hFcγRIIIb_NA2_ [91]. Omalizumab is currently approved for the treatment of moderate to severe persistent allergic asthma patients aged 6 years and older, CSU patients aged 12 years and older, and nasal polyp patients aged 18 years and older [92]. In 2018, the FDA granted a breakthrough therapy designation to omalizumab for food allergies as a monotherapy and combination therapy with OIT [93]. Though omalizumab was never approved for atopic dermatitis, an analysis of real-world data showed a clinical benefit of omalizumab in atopic dermatitis patients with low IgE levels [94]. However, most atopic dermatitis patients have high or extremely high IgE levels, where 75% of the patients have been reported to have IgE levels above 700 IU/mL, and 43% have IgE levels above 5000 IU/mL [94,95]. The dose of omalizumab is determined according to the patient’s weight and blood IgE levels [92], and for those whose IgE level is above a certain level for a specific body weight, there is not a recommended dosing, which means that the dose is arbitrarily determined by the physician. A recent phase 1b clinical trial with systemic lupus erythematosus (SLE) patients has shown that omalizumab is associated with improvement of the disease [96]. Although larger randomized clinical trials are needed to evaluate the efficacy of omalizumab in SLE patients, previous studies have shown that IgE autoantibodies may be involved in the development of SLE [97,98,99,100]. SLE patients have increased serum autoreactive IgE, and the presence of IgE autoantibodies is associated with increased basophil activation and enhanced disease activity, suggesting that IgE facilitates the amplification of autoimmune inflammation [98,99,100]. In *Lyn^−^*^/*−*^ mice, the activation of basophils by IgE autoantibodies amplifies autoantibody production, leading to lupus nephritis [97]. Similarly, in CSU patients with IgE autoantibodies against autoantigens such as thyroperoxidase (TPO) and double-stranded DNA, omalizumab relieves CSU symptoms by inhibiting IgE binding to FcεRI [101].

#### 4.1.2. Ligelizumab

Ligelizumab (QGE031), similar to omalizumab, is a humanized IgG1 mAb that recognizes the Cε3 domains of IgE but has a higher binding affinity to IgE than omalizumab [86,87]. Ligelizumab inhibits the binding of IgE to FcεRI more effectively than omalizumab. However, ligelizumab is less effective than omalizumab in suppressing the binding of IgE to the low-affinity IgE receptor CD23 (Table 1) [102]. Due to the fact that IgE binding to CD23 on B cells induces the inhibition of IgE production, ligelizumab is more effective than omalizumab in suppressing IgE production in B cells [102]. Unlike the expected, clinical trials of ligelizumab have not demonstrated a greater clinical benefit of ligelizumab over omalizumab. Ligelizumab showed better efficacy than omalizumab on inhaled and skin allergen provocation responses in patients with mild allergic asthma but not superior to the placebo or omalizumab in patients with severe asthma [103,104]. So, its development in asthma was eventually discontinued. Moreover, in a phase 2b clinical trial targeting CSU patients, ligelizumab showed better efficacy than omalizumab [105], which led to the breakthrough therapy designation of ligelizumab for CSU from the FDA [106]. However, in phase 3 clinical trials, ligelizumab failed to show superior efficacy to omalizumab, though it showed better efficacy than the placebo [107].

#### 4.1.3. UB-221

UB-221 is an IgG1 mAb (a humanized mouse 8D6 Ab) and its binding affinity (K_D_) to the Cε3 domain of IgE is 5.85 × 10^−11^ M, which is approximately a four times stronger affinity than omalizumab (K_D_ = 2.25 × 10^−10^ M) but about four times weaker than ligelizumab (K_D_ = 1.61 × 10^−11^ M) (Table 1) [108]. Unlike omalizumab and ligelizumab, UB-221 does not interfere with IgE binding to CD23 [108]. This allows UB-221 to effectively inhibit IgE synthesis in peripheral blood mononuclear cells (PBMCs) in an experimental setting where the IgE synthesis is stimulated by IL-4 and anti-CD40 Ab. UB-221 can simultaneously inhibit the IgE production of B cells and degranulation of effector cells such as mast cells (Table 1). When injected intravenously into cynomolgus monkeys, UB-221 reduced free IgE levels better than omalizumab [108]. UB-221 also reduced free IgE levels in the serum of patients with atopic dermatitis more effectively than omalizumab but at a similar level to ligelizumab [108]. In addition, a phase 1 study in CSU showed that the intravenous administration of UB-221 improved disease symptoms better than omalizumab by reducing free IgE levels [108].

#### 4.1.4. IgE_TRAP_

Presently, the fusion protein IgE_TRAP_ (GI-301), in which the IgD/IgG4 hybrid Fc domain is linked to the α chain of FcεRI, is being developed as an IgE blocking agent [37]. The α chain of FcεRI of IgE_TRAP_ has a high IgE affinity, about 69 times stronger than that of omalizumab (Table 1). In addition, the Fc domain of IgE_TRAP_ has no binding sites for FcγRs and complement component 1q (C1q), unlike the IgG1 Fc domain used for all other anti-IgE agents, and therefore, IgE_TRAP_ does not induce ADCC, complement-dependent cytotoxicity (CDC), or IgG-mediated anaphylaxis (Table 1) [37]. A combination approach with omalizumab has been sought to mitigate the potential risk of developing anaphylaxis to OIT. Considering that the use of omalizumab also has a potential risk of anaphylaxis [91], a non-IgG1 agent such as IgE_TRAP_ may be a better combination option for OIT. In fact, IgE_TRAP_ reduced free IgE levels in the sera of CSU patients more effectively than omalizumab in vitro and better controlled high blood IgE levels in cynomolgus monkeys when administered subcutaneously [37]. In an immunogenicity assay conducted by EpiScreen using CD8^+^-depleted PBMCs from at least 50 individuals with various HLA types, IgE_TRAP_ showed a T cell response in <10% of donors, comparable to the response observed from omalizumab and trastuzumab (anti-HER2 receptor mAb; Herceptin) used as negative controls [37]. At the corresponding level of response from the EpiScreen assay, omalizumab and trastuzumab showed clinical immunogenicity as low as 0.1% [109], which suggests that IgE_TRAP_ can also be expected to carry a low chance of immunogenicity. In addition, IgE_TRAP_ is a heavily glycosylated protein with seven sites for N-glycosylation in its FcεRIα extracellular domain and one site in its IgD/IgG4 hybrid Fc domain, which can be quickly cleared in vivo due to the interaction with glycan receptors. However, this problem was addressed by capping the glycan sites with sialic acid moieties, and increasing the sialic acid contents in IgE_TRAP_ suppressed its clearance and increased its bioactivity in vivo [37]. In a food allergy mouse model, a single injection of IgE_TRAP_ effectively inhibited allergic reactions in a dose-dependent manner. Interestingly, its therapeutic effect was enhanced by a combined treatment with probiotic *Bifidobacterium longum*, which was previously described to alleviate food allergy symptoms by secreting extracellular vesicles (EVs) that can specifically bind and induce mast cell apoptosis without affecting T cell immune response [37,110]. For instance, the injection of family 5 extracellular solute-binding protein, a main component of EVs, into mice markedly reduced the occurrence of diarrhea in a mouse food allergy model [110]. The combination of IgE_TRAP_ and *B. longum* inhibited free IgE levels and mast cell degranulation more effectively than IgE_TRAP_ alone and also induced a greater decrease in mast cell number and *Il-33* expression level in the intestine (Figure 2) [37]. These results highlight the potential benefit of adding probiotics to the IgE_TRAP_ therapy, especially in settings where treatment efficacy is suboptimal at the highest dosing allowed.

### 4.2. Inhibition of IgE-Independent Degranulation

Though no news on the development of MRGPRX2-targeting drugs is available to date, the involvement of MRGPRX2 in the IgE-independent mast cell degranulation pathway suggests MRGPRX2 as a potential target for anti-allergy drug development. One study showed the selective depletion of MRGPRX2^+^ mast cells in the skin using photosensitizer-conjugated anti-MRGPRX2 Ab injection and near-infrared irradiation [111]. Moreover, several studies reported the inhibition of MRGPRX2-mediated mast cell degranulation and the alleviation of inflammatory symptoms using various approaches, such as immunomodulatory single-strand oligonucleotide (ssON), DNA aptamer, chemicals, and tripeptide QWF [112,113,114,115]. These treatments inhibit mast cell degranulation by MRGPRX2, alleviating the inflammatory response. Recently, natural products such as flavonoids, phenols, triterpenoid saponins, chalcones, and glycosides also have been reported to inhibit MRGPRX2-mediated inflammatory responses [67].

In a rodent model of inflammatory bowel disease, the activation of P2RX7 induces intestinal inflammation through mast cells and causes enteric neuron death [78,116]. In patients with Crohn’s disease, P2RX7 expression is elevated in the colon [78]. P2RX7 antagonists such as brilliant blue G (BBG), A438079, and A740003 have shown beneficial effects in attenuating the inflammatory response in animal models [117,118]. AZD9056, a P2RX7 inhibitor, demonstrated beneficial effects in a phase 2a clinical trial with moderate to severe Crohn’s disease [119]. Currently, numerous efforts are ongoing to develop effective P2RX7 antagonists.

## 5. Conclusions

Mast cells act as central effector cells in IgE-dependent and IgE-independent inflammatory responses. Accordingly, mast cells are common targets for many new drugs in development for allergic diseases.

Earlier development programs were mostly anti-IgE Ab, focusing on the inhibition of IgE, and these efforts successfully translated into the approval of omalizumab as an allergy drug. In contrast, the strategy of targeting MRGPRX2 has yet to show any progress. More recent approaches to mast-cell-mediated diseases include eliminating or silencing mast cells, inhibiting mediators secreted by mast cells, and suppressing mast cell activation or signaling [120]. A better understanding of the biology behind mast cells concerning inflammation and their implication in allergic diseases provides a scientific basis for the development of more effective approaches to the treatment of allergies. Given the complexity in the pathology of allergic diseases, therapeutic strategies that combine treatment with a different mechanism of action are of value.

## Figures and Tables

**Figure 1 cells-12-01506-f001:**
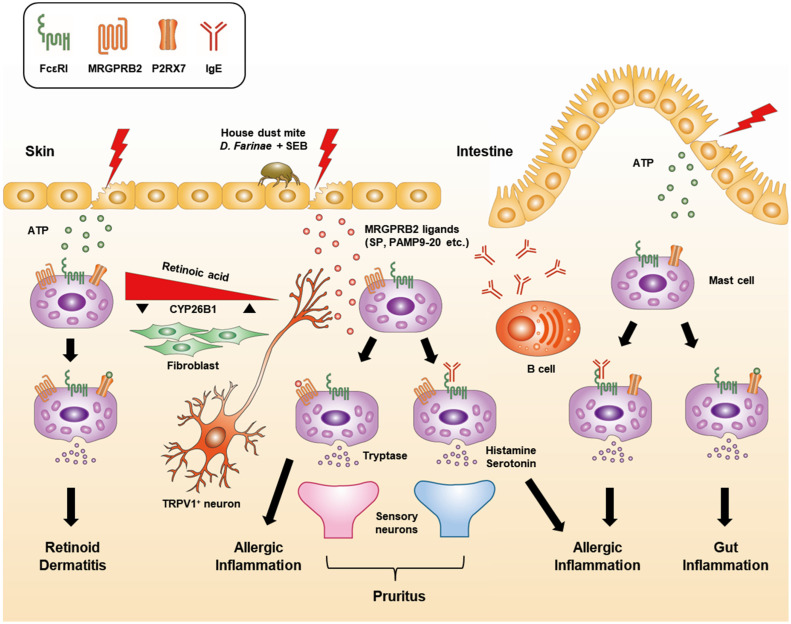
Inflammatory responses to IgE-dependent and -independent mast cell degranulation in the skin and intestine. FcεRI is constitutively expressed by mast cells in all tissues and plays a central role in allergic responses via its involvement in the IgE-mediated degranulation pathway. Skin mast cells can be distinguished from intestinal mast cells through their expression of MRGPRB2, which recognizes substance P (SP) secreted by transient receptor potential vanilloid type 1 (TRPV1^+^) neurons. Exposure to the house dust mite *Dermatophagoides farinae* and *staphylococcal* enterotoxins B (SEB) induces SP secretion, which activates mast cell degranulation through MRGPRB2 [72]. In addition, the activation of skin mast cells and the subsequent release of histamine and serotonin predominantly through the IgE-mediated degranulation pathway causes pruritus. In contrast, IgE-independent pruritis can be caused by the activation of mast cell degranulation by the MRGPRB2 ligand PAMP9-20 which results in tryptase-skewed secretion [75]. Therefore, MRGPRB2-mediated pruritus does not respond well to anti-histamine treatment. Unlike intestinal mast cells, skin mast cells do not express P2RX7 due to the high level of retinoic acid (RA)-degrading enzyme CYP26B1 in the fibroblasts and the RA concentration in the skin under homeostatic conditions [77]. However, the suppression of CYP26B1 activity or supplementation of excessive RA can induce P2RX7 expression by skin mast cells and trigger inflammatory responses to extracellular ATP produced by external stimuli.

**Figure 2 cells-12-01506-f002:**
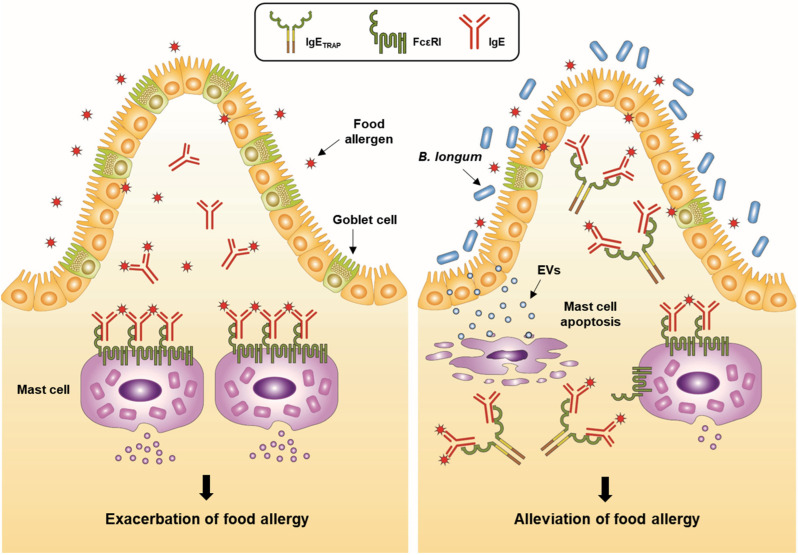
Suppression of food allergy by *Bifidobacterium longum* and IgE_TRAP_ combination therapy. Ingested food allergens allow the cross-linking of IgE with the high-affinity IgE receptor (FcεRI), which triggers the activation of mast cells and the release of effector molecules, marking the initiation of a type 2 hypersensitivity reaction. Goblet cell hyperplasia also occurs during the process. IgE_TRAP_ suppresses the activation and proliferation of mast cells by blocking the IgE from binding to FcεRI on mast cells [37]. Meanwhile, probiotic *B. longum* induces mast cell apoptosis through extracellular vesicle (EV) secretion, reducing mast cell numbers [110]. A strategic combination of treatments with a different target, such as probiotics with IgE_TRAP_, can improve clinical response through synergistic interactions between the treatments.

**Table 1 cells-12-01506-t001:** Comparison of properties of anti-IgE drugs.

	Omalizumab	Ligelizumab	UB-221	IgE_TRAP_
Drug form	Anti-IgE IgG1 Ab	Anti-IgE IgG1 Ab	Anti-IgE IgG1 Ab	FcεRIα–IgD/IgG4 Fc fusion protein
Binding domain of IgE	Cε3 domain	Cε3 domain	Cε3 domain	Cε3 domain and Cε2–Cε3 linker region
Fold increase in IgE affinity compared to omalizumab	1-fold	14-fold	4-fold	69-fold
Potential risk of low affinity FcγRs-associated side effects	+	+	+	−
Potential risk of C1q-associated side effects	+	+	+	−
Inhibition of CD23–IgE interaction	+	+	−	?
Inhibition of IgE production	−	−	+	?
Reference	[108]	[108]	[108]	[37,86]

+, positive response; −, negative response; ?, unknown.

## Data Availability

Not applicable.

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
