# Peer review of "Degranulation of Mast Cells as a Target for Drug Development"

_cells, 2023, doi:10.3390/cells12111506_

Round 1

Reviewer 1 Report

Overall this is a very interesting review. The review would benefit from grammatical revisions and removal of “the” before many of the nouns. For example, “the mast cells” and “the IL-4” is used in sentences where the “the” should be removed to improve grammatical correctness and readability. It would benefit from the addition of more content, noted below, for completeness.

-This sentence on line 101, “When administered to naive mice, IL-33 increases the expression of Th2 cytokines (e.g. IL-4, IL-5 and IL-13) in tissues and IgE levels in blood and causes eosinophilia [22].” needs to be reworked for correctness.

- ILC2 cells should be mentioned as the producers of il5 and il13 earlier in the paragraph or some edit needs to be done to make this sentence correct. The IgE increase needs to be related to IL-33 as a downstream effect, not a direct effect of IL-33. 

-Line 108 needs to clarify the type of allergic sensitization that is described in the manuscript. 

-In regards to mast cell produced IL-33, P2X7 needs to be mentioned in regards to intestinal mast cells and in the context of helminth infection. (doi: 10.1016/j.immuni.2017.04.017.)

-Line 171, “sever” should be “severe”

-In the Fcer1 section, the role of other Fc receptors should be mentioned or have an additional section. Fc gamma activating and inhibitory receptors are important for mast cells. Mouse anaphylaxis mediated through Fcgamma receptors should be mentioned.

-Additionally PAF receptor should be mentioned and considered.

-Section 4-1, CD23 is expressed on more cells than B cells, this needs to be edited.

-Age range that omalizumab is approved for by the FDA should be included. Additionally the expansion of omalizumab in various clinical trials to lupus should be discussed. Additionally, more should be added about omalizumab and CSU, as the mechanism of action is controversial.

-Only very few therapeutics methodology have been discussed. There are many more methodologies that were attempted and failed or stalled. For example, DARPins and Fcε-Fcγ fusion proteins. These therapeutic methods should be added for completeness of the review.

-In addition, discussion of vibration-induced mast cell degranulation and other mast cell degranulation methods should be summarized. doi: 10.1084/jem.20190701 as these receptors or ligands could be targeted therapeutically.

Author Response

<Point-by-point responses>

Reviewer 1

  1. Overall this is a very interesting review. The review would benefit from grammatical revisions and removal of “the” before many of the nouns. For example, “the mast cells” and “the IL-4” is used in sentences where the “the” should be removed to improve grammatical correctness and readability. It would benefit from the addition of more content, noted below, for completeness.

→ In accordance with the reviewer’s comment, we have tried to remove unnecessary “the” in front of nouns.

  1. This sentence on line 101, “When administered to naïve mice, IL-33 increases the expression of Th cytokines (e.g. IL-4, IL-5 and IL-13) in tissues and IgE levels in blood and causes eosinophilia [22].” Needs to be reworked for correctness.

ILC2 cells should be mentioned as the producers of il5 and il13 earlier in the paragraph or some edit needs to be done to make this sentence correct. The IgE increases needs to be related to IL-33 as a downstream effect, not a dirt effect of IL-33.

→ Correction has been made based on the reviewer’s comment and can be found on lines 117-123 of the revised manuscript.

  1. Line108 needs to clarify the type of allergic sensitization that is described in the manuscript.

→ According to the reviewer’s comment, the type of allergic sensitization has been clarified on line 130 of the revised manuscript.

  1. In regards to mast cell produced IL-33, P2X7 needs to be mentioned in regards to intestinal mast cells and in the context of helminth infection. (doi:10.1016/j.immuni.2017.04.017.)

→ After the reviewer’s suggestions, we added the section on P2X7 receptor in the revised manuscript.

  1. Lint 171, “sever” should be “severe”.

→ The typo noted by the reviewer has been corrected, which can be found on line 192 of the revised manuscript.

  1. In the FceRI section, the role of the Fc receptors should be mentioned or has an additional section. Fc gamma activating and inhibitory receptors are important for mast cells. Mouse anaphylaxis mediated through Fcgamma receptors should be mentioned.

→ As suggested by the reviewer, a section on FcgRs have been added, which can be found on lines 200-223 of the revised manuscript.

  1. Additionally, PAF receptor should be mentioned and considered.

→ Information on the PAF receptor can be found on lines 214-223 of the revised manuscript.

  1. Section4-1, CD23 is expressed on more cells than B cells, this needs to be edited.

→ Modification has been mad at the suggestion of the reviewer, which can be found on lines 333-336 of the revised manuscript.

  1. Age range that omalizumab is approved by the FDA should be included. Additionally, the expansion of omalizumab in various clinical trials to lupus should be discussed. Additionally, more should be added about omalizumab and CSU as the mechanism of action is controversial.

→ The age range for each of approved indications for omalizumab has been added, which can be found on lines 386-388 and 395-409 of the revised manuscript.

  1. Only very few therapeutics methodology have been discussed. There are many more methodologies that were attempted and failed or stalled. For example, DARPins and Fce-Fcg fusion proteins. These therapeutic methods should be added for completeness of the review.

→ After the reviewer’s comment, we described more methodologies attempted to block IgE. This section can be found on lines 339-375 of the revised manuscript.

  1. In addition, discussion of vibration-induced mast cell degranulation and other mast cell degranulation methods should be summarized. Doi:10.1084/jem.20190701 as these receptors or ligands could be targeted therapeutically.

→ The corresponding section has been added to lines 306-326 of the revised manuscript.

Reviewer 2 Report

The Review manuscript „Degranulation of mast cells as a target for drug development“ submitted to cells by Yang et al. allows the reader to gain valuable information on mast cells, degranulation-inducing receptors, and respective present and future drugs.

Nevertheless, certain points have to be added and corrected before the reviewer can support publication.

Major points:

- throughout (!) the manuscript, “c-Kit” has to be changed to “KIT”; the gene name is KIT (human) or Kit (mouse) and the protein name is only KIT for both human and mouse; protein names are written in capital letters only

- the same pertains to Mrgprb2/MRGPRB2 or MRGPRX2; the protein name is written in capital letters only

- chapter 2-1: some sentences on soluble SCF versus membrane SCF have to be included, particularly with respect to the differential mechanisms of signal transfer, paracrine versus juxtacrine (plus respective references)

- chapter 2-1: some sentences on synergistic mast cell activation by the combination of SCF and Ag have to be included (plus respective references)

- line 73: it has to be explained if the Il-4raF709 variant is expressed in mast cells or other cells or in all cells

- lines 183-186: the various agonists of MRGPRX2/B2 have to be specified

- Figure 1 + legend: MRGPRB2 has to be written in capital letters; “P2X7” has to be changed to “P2RX7” throughout the text; the same pertains to CYP26B1

- page 8: there should be more information on the content of references 29 and 79, since this is a very exciting topic

- Figure 2: this figure gives the impression that IgE first binds to the allergen and then the respective immune complex binds to the FceRI. The special ability of the FceRI, however, is its binding to the monomeric IgE; it is about affinity and not avidity; the figure should be changed accordingly or at least this has to discussed in the figure legend

- page 9: to complete the review, as chapter 4-3 some information on P2RX7 and potential inhibitors or treatments has to be given

Minor points:

- the last sentence of the abstract should be rephrased, particularly “…degranulation signal and tissue-specific expression…”

- line 79: “reconstructed” has to be changed to “reconstituted”

- line 82: “reconstruction” has to be changed to “reconstitution”

- lines 117-118: this sentence has to be rephrased since it is hard to understand

- line 131: “increasing mast cells” or “increasing mast cell numbers”? Has to be specified.

- line 171: “sever” has to changed to “severe”

- line 199: the gene name Tac1 has to be written in italics -> Tac1

- the same pertains to line 202: Mrgprb2Mut

- line 261: delete “of”

- lines 272-273: the relevant FcgR has to be concretized

- lines 282-283: please substantiate what this means for the treatment of respective patients

- lines 290-292: please rephrase; the reviewer thought that IgE binding to CD23 suppresses IgE production from B-cells, however this sentence suggests that IgE-binding by ligelizumab inhibits IgE production

- Table 1, left column: please specify the relevant FcgRs

- line 333: “CSU patients” instead of “CSU”

- line 337: the specificity of trastuzumab should be mentioned

- line 366: “MRGPRX2 Ab” has to be changed to “anti-MRGPRX2 Ab”

- line 369: “…tripeptide QWF inhibit…” should be changed to “…tripeptide QWF. These treatments inhibit…”

Author Response

<Point-by-point responses>

Reviewer 2

The Review manuscript ”Degranulation of mast cells as a target for drug development” to Cells by Yang et al. allows the reader to gain valuable information on mast cells, degranulation-inducing receptors, and respective present and future drugs.

Nevertheless, certain points have to be added and corrected before the review can support publication.

Major points:

  1. Throughout the manuscript, “c-Kit” has to be changed to “KIT”; the gene name is KIT (human) or Kit (mouse) and the protein name is only KIT for both human and mouse; protein names are written in capital letters only.

→ “c-Kit” has been corrected in the revised manuscript as the reviewer commented.

  1. The same pertains to Mrgprb2/MRGPRB2 or MRGPRX2; the protein name is written in capital letter only.

→ “Mrgprb2/MRGPRB2 or MRGPRX2” has been corrected in the revised manuscript as the reviewer commented.

  1. Chapter 2-1: some sentences on soluble SCF versus membrane SCF have to be included, particularly with respect to the differential mechanisms of signal transfer, paracrine versus juxtacrine (plus respective references).

→ This corresponding content has been added at the suggestion of the reviewer, which can be found on lines 56-71 of the revised manuscript.

  1. Chapter 2-1: some sentences on synergistic mast cell activation by the combination of SCF and Ag have to be included (plus respective references)

→ This content can be found on lines 67-71 of the revised manuscript.

  1. Line 73: it has to be explained if the Il-4raF709 variant is expressed in mast cells or other cells or in all cells.

→ A description of Il-4raF709 variant has been added on line 90-93 of the revised manuscript.

  1. Lines183-186: the various agonists of MRGRPX2/B2 have to be specified.

→ Additions to the reviewer’s suggestion can be found on lines 225-267 of the revised manuscript.

  1. Figure1+ legend: MRGRAPB2 has to be written in capital letters; “P2X7” has to be changed to “P2RX7” throughout the text; the same pertains to CYP26B1.

→ Figure 1 and legend have been also corrected as suggested by reviewer, and “P2X7” has been changed to “P2X7 receptor” throughout the revised manuscript.

  1. Page 8: there should be more information on the content of reference 29 and 79 since this is very exciting topic.

→ As suggested by the reviewer, the contents of references (29à38) and (79à113) have been added to lines 467-472 and 478-480 of the revised manuscript.

  1. Figure 2: this figure gives the impression that IgE first binds to the allergen and then the respective immune complex binds to the Fce The special ability of the FceRI, however, is its binding to the monomeric IgE; it is about affinity and not avidity; the figure should be changed accordingly or at least this has to discussed in the figure legend.

→ Figure 2 has been modified as suggested by the reviewer.

  1. Page 9: to complete the review, as chapter4-3 some information on P2RX7 and potential inhibitors or treatments has to be given.

→ The reviewer’s suggestion has been added to lines 511-518 of the revised manuscript.

Minor points:

  1. The last sentence of the abstract should be rephrased, particularly “…degranulation signal and tissue-specific expression…”

→ A correction as suggested by the reviewer can be found on line 16 of the revised manuscript.

  1. Line79: “reconstructed” has to be changed to “reconstituted”.

→ The typo noted by the reviewer has been corrected, which can be found on line 96 of the revised manuscript.

  1. Line 82: “reconstruction” has to be changed to “reconstitution”.

→ The typo noted by the reviewer has been corrected, which can be found on line 99 of the revised manuscript.

  1. Lines 117-118: this sentence has to be rephrased since it is hard to understand.

→ The revision made by the reviewer’s comments can be found on lines 138-140 of the revised manuscript.

  1. Line131: “increasing mast cells” or “increasing mast cell numbers” Has to be specified.

→ “increasing mast cells” has been changed to “increasing the intestinal mast cells numbers”, which can be found on line 153 of the revised manuscript.

  1. Line 171: “sever” has to be changed to “severe”.

→ The typo noted by the reviewer has been corrected, which can be found on line 192 of the revised manuscript.

  1. Line 199: the gene name Tac1 has to be written in italics.

→ “Tac1” has been corrected to “Tac1”, which can be found on line 249 of the revised manuscript.

  1. The same pertains to line202: Mrgprb2Mu

→ “Mrgprb2Mut” has been corrected to “Mrgprb2Mut”, which can be found on line 251 of the revised manuscript.

  1. Line 261: delete “of”

→ The typo noted by the reviewer has been corrected, which can be found on line 334 of the revised manuscript.

  1. Lines 272-273: the relevant FcgR has to be concretized.

→ As suggested by the reviewer, specific FcgRs have been included, which can be found on lines 385-386 of the revised manuscript.

  1. Lines 282-283: please substantiate what means for the treatment of respective patients.

→ As suggested by reviewer, the content has been added in the lines 395-398 of the revised manuscript.

  1. Lines 290-292: please rephrase; the reviewer though that IgE binding to CD23 suppression IgE production from B cells, however this sentence suggests that IgE-binding by ligelizumab inhibits IgE production.

→ In accordance with the reviewer’s comment, the text has been corrected to prevent reader misunderstanding, which can be found on lines 416-418 of the revised manuscript.

  1. Table 1, left column: please specify the relevant FgcRs.

→ Following the reviewer’s suggestion, the FcgRs in the left column of Table 1 have been specified.

  1. Line 333: “CSU patient” instead of “CSU”

→ It has been corrected according to the reviewer’s comment, which can been found on line 459 of the revised manuscript.

  1. Line 337: the specificity of trastuzumab should be mentioned.

→ It has been specified according to the reviewer’s comment, which can been found on line 464 of the revised manuscript.

  1. Line 366: “MRGPRX2 Ab” has to be changed to “anti-MRGPRX2 Ab”

→ It has been corrected according to the reviewer’s comment, which can been found on line 503 of the revised manuscript.

  1. Line 369: “… tripeptide QWF inhibit…” should be changed to “…tripeptide QWF. These treatment inhibit…”

→ It has been corrected according to the reviewer’s comment, which can been found on line 507 of the revised manuscript.

Round 2

Reviewer 1 Report

The revisions are completed up to standards.

Author Response

Reviewer 1

The revisions are completed up to standards.
→ There were no additional comments.

Reviewer 2 Report

The manuscript definitely was improved by the changes added by the authors. However, certain points have still to be corrected before publication can be advised.

- line 12: change "P2X7" to "P2RX7"

- line 65: change "Kit" to "KIT"; the same pertains for lines 156 and 157 

- line 141: what kind of protein is B2X7? Do the authors mean P2X7? If yes, then it has to be changed

- "P2X7 receptor" has to be changed to "P2RX7"; there is no receptor for P2X7, which would be the meaning of "P2X7 receptor"; this has to be changed throughout the text (approximately 10 times or even more) as well as in Figure 1

- line 318: the last word "is" has to be deleted

- line 321: "founded" ha to be changed to "found"

- line 353: the last word has to be changed to "increased"

Author Response

Reviewer 2

The manuscript definitely was improved by the changes added by the authors. However, certain points have still to be corrected before publication can be advised.

  1. line 12: change "P2X7" to "P2RX7"

→ “P2X7” has been changed to “P2RX7” on line 12 of the revised manuscript.

  1. line 65: change "Kit" to "KIT"; the same pertains for lines 156 and 157

→ In accordance with the reviewer’s comment, “Kit” has been corrected to “KIT” on lines 65, 157 and 158 of the revised manuscript.

  1. line 141: what kind of protein is B2X7? Do the authors mean P2X7? If yes, then it has to be changed.

→ As the reviewer said, B2X7 is typing mistake of P2X7. “B2X7 receptor” has been corrected to “P2RX7” on line 142 of the revised manuscript.

  1. "P2X7 receptor" has to be changed to "P2RX7"; there is no receptor for P2X7, which would be the meaning of "P2X7 receptor"; this has to be changed throughout the text (approximately 10 times or even more) as well as in Figure 1

→ As per the reviewer’s comment, ”P2X7 receptor” has been changed to “P2RX7” throughout the revised manuscript (including Figure 1).

  1. line 318: the last word "is" has to be deleted

→ The last word “is” has been deleted on line 320 of the revised manuscript.

  1. line 321: "founded" has to be changed to "found"

→ “founded” has been corrected to “found” on line 323 of the revised manuscript.

  1. line 353: the last word has to be changed to "increased".

→ The last word “increase” has been changed to “increased” on line 355 of the revised manuscript.
